# A Vaccine Construction against COVID-19-Associated Mucormycosis Contrived with Immunoinformatics-Based Scavenging of Potential Mucoralean Epitopes

**DOI:** 10.3390/vaccines10050664

**Published:** 2022-04-22

**Authors:** Muhammad Naveed, Urooj Ali, Mohmed Isaqali Karobari, Naveed Ahmed, Roshan Noor Mohamed, Shahabe Saquib Abullais, Mohammed Abdul Kader, Anand Marya, Pietro Messina, Giuseppe Alessandro Scardina

**Affiliations:** 1Department of Biotechnology, Faculty of Life Sciences, University of Central Punjab, Lahore 54000, Pakistan; dr.naveed@ucp.edu.pk (M.N.); msurooj9@gmail.com (U.A.); 2Center for Transdisciplinary Research (CFTR), Saveetha Institute of Medical and Technical Sciences, Saveetha Dental College, Saveetha University, Chennai 600077, India; 3Department of Restorative Dentistry & Endodontics, Faculty of Dentistry, University of Puthisastra, Phnom Penh 12211, Cambodia; 4Department of Medical Microbiology and Parasitology, School of Medical Sciences, Universiti Sains Malaysia, Kubang Kerian 16150, Malaysia; namalik288@gmail.com; 5Department of Pediatric Dentistry, Faculty of Dentistry, Taif University, P.O. Box 11099, Taif 21944, Saudi Arabia; roshan.noor@tudent.edu.sa; 6Department of Periodontics and Community Dental Sciences, College of Dentistry, King Khalid University, Abha 61421, Saudi Arabia; drsaquib24@gmail.com; 7Department Restorative Dental Science, College of Dentistry, King Khalid University, Abha 61421, Saudi Arabia; msaheb@kku.edu.sa; 8Department of Orthodontics, University of Puthisastra, Phnom Penh 12211, Cambodia; amarya@puthisastra.edu.kh; 9Department of Surgical, Oncological and Stomatological Disciplines, University of Palermo, 90133 Palermo, Italy; pietro.messina01@unipa.it

**Keywords:** mucormycosis, immunoinformatics, vaccine design, vaccine efficacy, population coverage, immune activation

## Abstract

Mucormycosis is a group of infections, caused by multiple fungal species, which affect many human organs and is lethal in immunocompromised patients. During the COVID-19 pandemic, the current wave of mucormycosis is a challenge to medical professionals as its effects are multiplied because of the severity of COVID-19 infection. The variant of concern, Omicron, has been linked to fatal mucormycosis infections in the US and Asia. Consequently, current postdiagnostic treatments of mucormycosis have been rendered unsatisfactory. In this hour of need, a preinfection cure is needed that may prevent lethal infections in immunocompromised individuals. This study proposes a potential vaccine construct targeting mucor and rhizopus species responsible for mucormycosis infections, providing immunoprotection to immunocompromised patients. The vaccine construct, with an antigenicity score of 0.75 covering, on average, 92–98% of the world population, was designed using an immunoinformatics approach. Molecular interactions with major histocompatibility complex-1 (MHC-I), Toll-like receptors-2 (TLR2), and glucose-regulated protein 78 (GRP78), with scores of −896.0, −948.4, and −925.0, respectively, demonstrated its potential to bind with the human immune receptors. It elicited a strong predicted innate and adaptive immune response in the form of helper T (Th) cells, cytotoxic T (TC) cells, B cells, natural killer (NK) cells, and macrophages. The vaccine cloned in the pBR322 vector showed positive amplification, further solidifying its stability and potential. The proposed construct holds a promising approach as the first step towards an antimucormycosis vaccine and may contribute to minimizing postdiagnostic burdens and failures.

## 1. Introduction

Mucormycosis, caused by fungal infiltration of blood vessels, is linked to a high mortality rate. This angioinvasive infection is acquired predominantly by sporangiospore inhalation, but the chances of infection through traumatic, nonsterile inoculation and contaminated food are equally high [1,2]. Hematological malignancy, diabetes mellitus, corticosteroid treatment, exposure to skin burns to the environment, and solid organ transplants increase the risk of mucormycosis [3]. Mucormycosis has plagued the world for 5 decades, but little is known about its causative agents, pathogenesis, and epidemiology [4]. Therefore, therapeutic approaches to treat this problem are also scarce [5]. During the COVID-19 pandemic, India has had the highest number of cases of mucormycosis among COVID-19 patients, which was almost an 80 times higher rate compared with the rest of the world [6].

As expected, the risk factors for mucormycosis also differ. Diabetes mellitus (DM) has been reported as the most prevalent risk factor for mucormycosis in Asian countries, whereas transplantation and malignancies occupy this spot in the European world [6,7]. Apart from these general risk factors, research in the past 2 decades has brought many health-care-associated cases of mucormycosis to light [8]. Its incidence is constantly increasing with India and China being the most hit countries. Diabetes remains the most common risk factor in these cases, especially in India, where DM and mucormycosis top the charts in their respective incidence rates compared with relevant health problems, respectively [9]. Along with DM, studies have associated corticosteroid use and abuse with mucormycosis. Immunocompromised individuals, if provided 2–7 g methyl prednisone doses, are naturally predisposed to be attacked by mucormycosis-causing agents [10].

The most prevalent type of mucormycosis is rhino-orbito-cerebral mucormycosis (ROCM), which is frequently observed in individuals with ketoacidosis or uncontrolled diabetes mellitus. The second most obvious and currently the most predominant clinical form of mucormycosis is the pulmonary (PM) type. It is most often seen in patients with transplants and hematological issues. COVID-19 is another major contributor to the escalated development of PM [11]. Among its risk factors, blood malignancy is said to be the major factor with a contribution of 32–40% [12]. Diabetes comes second, followed by other factors, such as renal diseases, hematopoietic stem cell, and organ transplants. High fever, pleuritic chest pain, consistent cough, and dyspnea are among its symptoms [13].

Mucormycosis has a wide range of clinical manifestations, based on the host’s pre-existing immunosuppression. Even though there is substantial diversity, the clinical manifestation may be generally classified into six categories, ROCM [4], pulmonary [14], gastrointestinal, cutaneous [15], disseminated, and renal mucormycosis [16], based on anatomic predilection. Among its causative agents, *Rhizopus* organisms are said to be behind ROCM, *Saksenaea* and *Apophysomyces* are usually identified in cutaneous mucormycosis, and *Cunninghamella* organisms are usually associated with disseminated and pulmonary forms of the infection [6].

Despite a redundant and fatal attack of mucormycosis every now and then, therapeutics are focused on postdiagnosis solutions. In most cases, it gets too late to apply that solution. Ibrahim et al. (2010) worked on developing an Ftr1 vaccine to counter mucormycosis; however, that project was limited to the demonstration of a potential antibody vaccine [17]. Today, when the world has been shaken by the disease again, mucormycosis has threatened to become a global concern. This demands a rigorous analysis of the available therapeutic approaches against this deadly chain of fungal infections and an understanding that a vaccine may minimize the chances of mucormycosis spread and fatalities.

This study proposes a multiepitope vaccine against mucormycosis. *Mucor* and *Rhizopus* species are known to be at the heart of a majority of mucormycosis infections, so it targets one transmembrane protein from each species and combines their antigenic epitopes to construct a multipotent vaccine that can protect the human host from both species’ infections. It will consider the global coverage of epitopes selected, immune stimulation, and response to the vaccine, and will provide a cloning insert to help future researchers with the in vitro vaccine development. Although performing a rigorous, extensive, and thorough analysis, we have amalgamated epitopes from proteins of two species to construct the vaccine, which can be improved since mucormycosis is a broad-spectrum disease and epitopes from other species could also be of significance. Furthermore, it is solely a computational approach and lacks the sensitivity of dry lab assessments. For an efficient vaccine, molecular and immunological assays can contribute a great deal.

## 2. Results

### 2.1. Target Proteins’ Selection

Upon analyzing 6000 proteins for *R. oryzae* and *M. circinelloides* from CELLO2GO, 18 proteins for both organisms were short-listed as they were present in the transmembrane region. Out of these 18, phytoene dehydrogenase from *M. circinelloides* (UniProt ID: tr|Q9Y798) and a hypothetical protein from *R. oryzae* (accession number: KAG1630349.1) were selected for vaccine design. Both proteins showed a hundred percent conservation among various rhizopus and mucor species. The selected proteins were found to be antigenic and nonallergenic, which made them fit for further analyses. Table 1 provides the antigenicity scores of the selected proteins.

### 2.2. B-Cell Epitope Prediction

The BepiPred 2.0 tool predicted B-cell-specific epitopes for both the proteins. On average, the predicted score for the epitopes of phytoene dehydrogenase fitness was 0.450 with a minimum of 0.218 and a maximum score of 0.663. For the hypothetical protein, the average score was 0.444 with a minimum of 0.213 and a maximum score of 0.690. According to preferential surface accessibility, 13 peptides for phytoene dehydrogenase and 6 peptides for the hypothetical protein were selected. The average score of accessibility for both the phytoene dehydrogenase and the hypothetical protein residues was 1.000; the graphs for surface accessibility and antigenicity of both the proteins are provided in Figure 1.

The analysis provided 20 peptides with antigenicity above the average and a threshold value of 1.027 with the minimum score being 0.889 for the residues of phytoene dehydrogenase (Figure 1A). This prediction was fascinating as the majority of the peptides demonstrated an antigenic potential. The results were duplicated for the hypothetical protein residues, which averaged an antigenicity score of 1.014 with the maximum score being 0.874 and the maximum being 1.164, provided in Figure 1B. The flexibility, β-turn, and hydrophilicity analyses predicted 416 epitopes based on flexibility and 576 epitopes based on β-turn and hydrophilicity, respectively, for the phytoene dehydrogenase residues. Fifty percent of phytoene dehydrogenase residues in all the graphs were satisfying the threshold values. For the hypothetical protein, 281 residues based on flexibility and 282 based on β-turn and hydrophilicity were calculated. According to surface accessibility, which was selected as an important factor for analyses, based on predictions of epitopes that are easily accessible (present) on the surface of the target proteins, 13 peptides from phytoene dehydrogenase and 6 for the hypothetical protein were provided, with the average scores being 1.000 for both proteins (Figure 1C,D).

### 2.3. Protein Modeling, Refinement, and Validation

trRosetta predicted the tertiary protein structure for phytoene dehydrogenase with a Tm score of 0.846 and the structure for hypothetical protein with a Tm score of 0.742. The models are demonstrated in Figure 2C,D along with the GalaxyRefine structures of phytoene dehydrogenase and hypothetical protein respectively. For phytoene dehydrogenase, ERRAT predicted that the overall protein quality was 95.2212. For the hypothetical protein, ERRAT showed a quality factor of 84.838, which was well above the accepted value of 50. PROCHECK predicted that 95.1% of the residues of the phytoene dehydrogenase protein lie in the favorable region according to the RC plot. Zero bad contacts were predicted, whereas 99% of the planar groups were within limits.

PROCHECK predicted that 89.8% of the hypothetical protein’s residues fall in the core category with 9.0% in the allowed category. Only 1 bad contact was found in the protein, and 99% of planar groups were within limits. According to the RC plot, 95.053% residues were found to be in the highly preferable regions, and 4.240% residues were shown in additionally allowed regions, whereas 0.7% residues were present in the questionable category. Verify3D scores predicted that 97.75% phytoene dehydrogenase residues averaged >0.2 scores. Additionally, a minimum of 80% of AAs passed the 3D/1D profile complementing the overall score. Similarly, for the hypothetical protein, 90.53% scored >0.2, and more than 80% of residues passed the 3D/1D scores as well.

### 2.4. Discontinuous B-Cell Epitope Analysis

ElliPro predicted 15 linear epitopes and 8 discontinuous epitopes for phytoene dehydrogenase along with 11 linear and 3 discontinuous epitopes for hypothetical protein. The discontinuous epitopes for phytoene dehydrogenase and hypothetical protein (shown in Figure 2A,B) are provided in Appendix A.

### 2.5. Selection of B-Cell Epitopes

Out of the 20 epitopes retrieved from the antigenicity analysis and 13 epitopes retrieved from the surface accessibility analysis for phytoene dehydrogenase, 3 epitopes (SVIVLVPIG, KMVLAVIER, and ILGLSHDVLQVLWF) were selected for the final construct on the basis of their antigenicity, accessibility, and allergenicity values. Similarly, out of the 6 surface accessibility epitopes and 10 antigenicity-based epitopes for the hypothetical protein, 3 (DKIYKKTTKH, VLTHVDLIEKLLHYN, and IQLISPPSKKSKTT) were finalized for the final vaccine construct. A total of 6 B-cell epitopes were selected. The antigenicity and accessibility scores for these residues are provided in Table 2.

### 2.6. T-Cell Epitope Analysis

For phytoene dehydrogenase, 15,738 interactions with all the HLA molecules were predicted, out of which 50 epitopes’ IC50 values were below the threshold, and hence, the remaining results were discarded. Out of these 50, 10 were further processed for population coverage analysis. For the hypothetical protein, 12,584 interactions were predicted, out of which 23 epitopes’ IC50 values were below the threshold value of 500, and 6 were further run for population coverage.

Among phytoene dehydrogenase’s epitopes, MAFTFQTMY was a dominant binder to 10 HLA alleles (HLA-C*07:01, HLA-B*15:01, HLA-A*03:01, HLA-A*11:01, HLA-B*53:01, HLA-A*30:02, HLA-B*58:01, HLA-A*68:01, HLA-B*57:01, HLA-B*35:01), WVMFMFFYF and LTSSISFY were dominant binders of 9 alleles (HLA-A*32:01, HLA-A*68:01, HLA-A*24:02, HLA-A*68:02, HLA-B*53:01, HLA-B*15:01, HLA-B*35:01, HLA-A*23:01, HLA-A*02:06 and HLA-B*35:01, HLA-A*03:01, HLA-B*58:01, HLA-A*26:01, HLA-A*11:01, HLA-B*15:01, HLA-A*68:01, HLA-A*30:02, HLA-A*01:01), respectively. Among hypothetical protein’s six epitopes, QMFNPPFVY and RLMNGHNSM were dominant binders of 10 HLA alleles, STIDPAQSY bound 9 HLA alleles, RYCCRRMVL and KVYEWDFSR bound 6 HLA alleles dominantly, and YLSLIQAEY bound to 5 HLA alleles.

MHC-II analyses provided quite an extensive result wherein out of 44,523 allelic interactions, 10 core peptides of the phytoene dehydrogenase were chosen with all the epitopes having an IC50 value of <100. Hypothetical protein’s results were surprisingly elaborative with one of the core peptides (FVYSLAIST) binding dominantly to 177 HLA-DR alleles with IC50 values of <100. A total of 14 core peptides were selected.

### 2.7. Population Coverage

Selected MHC-I epitopes showed phenomenal accumulative world coverage of 97.15% with a PC90 value of 3.08, whereas the MHC-II epitopes displayed a coverage of 77.71% coverage, with a PC90 average of 0.45, which suggests the minimum epitopes hit per HLA combination acknowledged by 90% of the population. This indicated the potential of the selected epitopes for the final vaccine construct. The coverage maps of phytoene dehydrogenase’s MHC-I and MHC-II epitopes are shown in Figure 3A,B, respectively. The former suggested that epitope 4 covered the highest percentage of the world population, estimated to be 22% on average, whereas the latter suggested the same about the first epitope with an estimated population coverage of 23%.

The combined world coverage displayed extraordinary results with a 98.26% coverage of the selected epitopes of phytoene dehydrogenase (Figure 3C). The average hit per HLA antigen was estimated to be 3.75, whereas the PC90 value was calculated to be 1.86. The coverage for individual epitopes for MHC-I and MHC-II classes are provided in Table 3, and the graphical coverage is provided in Figure 3 (A: individual MHC-I coverage of the epitopes, B: individual MHC-II coverage of the epitopes, C: combined population coverage of both the MHC-I and MHC-II epitopes). The graph (Figure 3C) predicted that epitope 4 of the phytoene dehydrogenase, MAFTFQTMY, covered the highest percentage of individuals, estimated to be 62.52%. The world population coverages of the hypothetical protein’s epitopes were not as astounding as the ones for phytoene dehydrogenase; however, they were quite satisfactory. For the MHC-I class, the epitopes covered 90.78% of the population with an average 3.01 hit and a 1.05 PC90 value (Figure 3D). For the MHC-II class, the core epitopes covered 58.33% of the world population with an average hit of 0.98 and a PC90 value of 0.24 (Figure 3E). The combined coverage of all the epitopes for both classes was, however, excellent, covering 96.16% of the world population, with an average hit of 3.99 and a PC90 value of 1.62 (Figure 3F). Individual coverage for each epitope against MHC-I and MHC-II and combined coverage are provided in Table 3. QMFNPPFVY, RYCCRRMVL, IDLNESNKF, and STIDPAQSY showed the best individual coverages of 66.02%, 49.95%, 33.99%, and 45.26%, respectively.

### 2.8. Selection of T-Cell Epitopes

Out of 10 MHC-I and MHC-II epitopes of phytoene dehydrogenase, 6 core epitopes were nonallergenic as per AllerTOP v.2.0. Out of 5 MHC-I and MHC-II epitopes of the hypothetical protein, 4 epitopes and 3 core peptides were considered nonallergenic. Among the selected 6 core epitopes of phytoene dehydrogenase and 3 core epitopes of the hypothetical protein, 2 for each satisfied the VaxiJen v.20 analysis at a threshold of 0.5, and 6 epitopes from phytoene dehydrogenase and 8 from the hypothetical protein were finalized for vaccine construct based on their antigenicity scores, as shown in Table 4.

### 2.9. Final Vaccine Construct

Figure 4A depicts the final vaccine construct using the selected B- and T-cell epitopes. The green sequence indicates the adjuvant, the red ones are linkers, and the golden sequence indicates the polyhistidine tags that stabilize the protein structure and protect it from nuclease action.

### 2.10. Vaccine Antigenicity and Allergenicity

Table 5 provides the antigenicity and allergenicity analyses of the vaccine construct. ToxinPred provided superlative results, demonstrating that only 8 residues of the construct were toxic, and overall, the construct was nontoxic by a huge margin. A partial prediction of ToxinPred along with the toxin residues, highlighted in red, is provided in Figure 4B.

### 2.11. Prediction of Secondary Structure

The software indicated the vaccine to be a transmembrane protein, validating its origin from two transmembrane proteins. PSIPRED predicted that most of the vaccine was involved in a transmembrane helix and only 31 residues (7%) were disordered. The MEMSTAT analysis predicted that residues from 314 to 458 are extracellular, exposed to the host proteins. Figure 5 provides the PSIPRED and MEMSTAT analyses. Appendix A presents the disordered plot of the vaccine. Only the residues crossing the cutoff value (0.5) were said to be disordered. It was not problematic because only 3 of the 31 residues were from the primary construct, and the rest were part of the adjuvant or polyhistidine tags. Scratch provided similar results shown in Appendix A. Only 1 disulfide bond was predicted in the structure between the cysteines present on positions 319 and 320. The antigenicity score predicted by Scratch, as shown in Table 5, was 0.5, indicating that the vaccine construct is antigenic.

### 2.12. Tertiary Structure Prediction, Refinement, and Verification

trRosetta predicted the tertiary structure of the vaccine with a TM score of 0.237. Such a low score indicates the uniqueness of the protein structure. The contact maps of the predicted 2D structure indicate the credibility of the protein structure. ERRAT predicted that the overall protein quality was 88.452, as shown in Figure 6A. PROCHECK predicted that 94.9% of the residues of the vaccine construct lie in the favorable region according to the RC plot, as shown in Figure 6B.

The RC plot suggested that right-handed helices and β-sheets are the most dominant secondary structures in the protein corresponding to good hydrophilicity. Only 0.5% of residues were in the disallowed region, which proved the protein quality. Verify3D scores predicted that 67.90% of the construct’s residues averaged >0.2 scores. Additionally, a minimum of 80% of AAs passed the 3D/1D profile, complementing the overall score. The overall quality of the predicted 2D structure was satisfactory with most of the residues concentrated in the lower left of the graph, indicating the presence of α-helices.

### 2.13. Physicochemical Properties

The number of amino acids submitted to ProtParam was 458. The output molecular weight was 49051.19 g/mol. The adjuvant part was considered as the N-terminal of the vaccine, and the estimated half-life in reticulocytes according to in vitro analysis was 30 h, which is a quality attribute. The instability index was calculated to be 34.96, whereby the threshold value is 40. This prediction indicated that the vaccine construct is stable. The aliphatic index predicted by the server was 78.52. The GRAVY index was calculated to be −0.064, indicating the hydrophilicity of the vaccine construct. Solubility analysis predicted by the Protein-Sol also predicted the same about vaccine hydrophilicity or solubility. The index is given in Figure 6C, whereas the folding propensity and charge score per amino acid is provided in Figure 6E.

### 2.14. ElliPro and Cleavage Analysis

These analyses showed that majority of the epitopes, including MAKLSTDEL, STIDPAQSY, SIDLNES, YSLAIST, and DGNWIAAYDKIY, were conserved after fusing them into a new construct. Figure 6D demonstrates the 3D discontinuous epitope processed by ElliPro with a score of 0.801 and 57 residues.

### 2.15. Molecular Docking

Docking analyses of vaccines with MHC-I gave 25 clusters with varying energies, out of which model 0 was selected for simulations. It had the lowest energy, −987.5, and its center was estimated to be −896.0. The docking complex is provided in Figure 7A, wherein blue and green represent the MHC-I molecule and orange, red, and yellow are the vaccine. Figure 7B shows the docking complex of TLR2 with the vaccine, wherein blue and green represent the receptor. Out of 29 provided clusters by ClusPro, model 25 was selected based on the lowest energy and the centers being −948.4. Lastly, the docking complex of the vaccine with the natural mucormycosis receptor is shown in Figure 7C. A total of 28 complexes were provided by the server, and model 1 was selected based on its score, −925.0, and center, −889.8.

A total of 48, 55, and 26 interactions between the ligand and the receptors, MHC-I, TLR2, and GRP78, respectively, were computed from Discovery Studio. MD simulations helped visualize the molecular interactions, as shown in Figure 8A, for the docking complex of GRP78 and vaccine. Figure 8B provides the improved simulation complex of MHC-I and vaccine. Figure 8C provides similar results for TLR2 and vaccine.

### 2.16. Codon Optimization

The reverse-translated sequence processed by EMBOSS, when run on JCAT, provided sublime optimization with a CAI value of 1.0, indicating that the refined codons will show a maximum expression in the host, and GC contents reduced from 61% to 50%. Figure 9A,B provides the codons after optimization and the optimized sequence, respectively. The red lines in both figures represent the codon adaptation. Comparing the preadaptation sequence and the post adaptation sequence, we found an idea that C is replaced by T in the fifth position, changing the codon from GCC to GCT. However, both these codons translate to the same amino acid, alanine, hence not changing the final protein sequence but optimizing the codon according to the host.

### 2.17. Cloning and Expression

Cloning was performed by replacing the EcoRV and Nru1 restriction sites with our vaccine construct (Figure 9C) using the direct insert method. The precloning construct was of 4361 base pairs, whereas after insertion, the vector size increased to 4948 base pairs. The EcoRV restriction site was chosen because it was also present in the vaccine construct and enhanced the chances of stable insertion.

### 2.18. Immune Response Simulation

C-ImmSim provided an excellent immune response to the designed vaccine. Figure 10 provides the B-cell population: (A) PLB population, (B) B-cell presentation, (C) and Th cell population (D) per subsequent injection day. According to these graphs, active B cells, memory B cells, and IgM isotypes are constitutively produced and duplicated over the course of the vaccine. The PLB graph depicts the exponential growth of IgM + IgG isotypes and individual IgM and IgG1 isotypes following the 1st week of vaccine injection. The B-cell population/presentation graph depicts the active presentation of MHC-II molecules in the first 5 days of injection. Additionally, B cells stay active for a long time after the injection, which indicates the efficacy and quality of the proposed vaccine. Lastly, the helper T-cell graph shows that the memory cells and antigen-presenting T cells stay active for a long course, whereas Th cells’ concentration slightly decreases after 15 days of injection.

Th0 activity decreases as Th1 activity increases in response to the vaccine. The memory TC cells are produced constantly during the 1st month of injection, while the concentration of active TC cells decreases after the first 2 weeks. The NK cell population showed a pulsating increase and decrease; however, during the first 10 days of injection, their population was quite extraordinary. The MA graph per state depicts both the active and resting populations increasing in the first 30 days of injection. Antigen presentation increases in the first 3 days and then decreases spontaneously.

## 3. Discussion

With the increasing threat of mucormycosis adjoined with COVID-19 variants and underfacilitated hospital systems, a permanent intervention is required. As mentioned by Imran et al. (2021), mucormycosis, although dangerous and lethal for immunocompromised people, is equally infectious to immunocompetent people, and no vaccine has yet been developed to minimize its chances of infection [18]. This study provided an immunoinformatics approach to bridge this gap and propose a vaccine construct from two highly infectious species of Mucorales responsible for the majority of mucormycosis infections worldwide. Structural and computational analyses provide a benchmark for proposing therapeutics that are not cost-effective in the preliminary phases of design. Gupta and Kumar (2020) utilized a similar approach to propose a vaccine against a variety of *C. jejuni* [19]. Furthermore, Elamin Elhasan et al. (2021) also worked on a similar approach and proposed a multiepitope vaccine against *Candida glabrata* [20].

These, along with other innovations, support the potential use of bioinformatics, computational vaccinology, and immunoinformatics in the fields of health and medicine. Chaudhuri and Ramachandran (2017) consider immunoinformatics as a golden weapon in applying techniques such as reverse vaccinology [21]. The efficacy of vaccines proposed computationally is proved by a lot of recent efforts. Following the lead, this study utilized 6 B-cell and 14 T-cell epitopes to construct a vaccine. Despite an average coverage, the selected epitopes showed phenomenal immune simulations, establishing that the antigenicity of the selected epitopes plays a major role in an efficient drug design. Most of the work for the study was processed on IEDB tools, and the application was well suited and enough for thorough analysis, as indicated by Dhanda et al. (2019) [22]. A variety of software were utilized to verify the results, and fortunately, all the analyses were consistent with one another.

A recently published study identified 80 COVID-19-associated mucormycosis [23]. Among these cases, 42 cases were reported from India; 8 from the USA; 5 from Pakistan; 4 from France, Iran, and Mexico; 2 from Russia; and 1 from Bangladesh, Austria, Brazil, Chile, Germany, Kuwait, Italy, Lebanon, the UK, Turkey, and Czech Republic [23]. Most of these cases were reported when the patients were being treated at respective health-care centers, while some of the cases were reported after the patients had recovered from COVID-19. The most common types of mucormycosis reported were ROCD and pulmonary mucormycosis [23].

Roden et al. (2005) reported that the majority of mucormycosis episodes were associated with *Rhizopus* species, found in 47% of the cases, followed by *Mucor*, found in 18% of the cases, in a survey of over 900 documented human reports [24]. The genera *Mucor, Rhizopus,* and *Lichtheimia* are most reported, among which *Rhizopus arrhizus* is the most dominant organism [6]. Although these are the most common agents, infections due to *Cunninghamella* are said to be the most fatal, with only 7% pathogenesis recorded [24]. Other species involved in the pathogenesis are *Apophysomyces*, *Absidia* species, *Saksenaea species*, and *Rhizomucor pusillus* with 5%, 5%, 5%, and 4% documentation, respectively.

An important aspect of designing any vaccine construct is to ensure its antigenicity, nonallergenicity, and nontoxicity. Results for all three aspects were consistent with that of Naveed et al. (2021), who worked on antigenicity analyses throughout [25]. This study utilized a similar approach, and the individual antigenicity and allergenicity of all the epitopes were processed for the selection of final epitopes. To our knowledge, this practice ensured the final antigenicity of our vaccine to elicit the required immune response. The choice of adjuvants and linkers was a fascinating aspect of this study. After searching for a lot of fungal adjuvants and linkers, the results obtained were surprisingly not satisfactory. Tarang et al. (2020) used a TLR4 agonist to enhance the fungal vaccine activity; however, this adjuvant destabilized our protein structure [26]. Utilizing the adjuvant used by Naveed et al. (2021) helped us achieve the desired stability and did not affect the antigenicity [25]. The reason why the adjuvant used by Tarang et al. (2021) did not work is not yet understood [26].

Furthermore, the use of C-ImmSim eased the process of immune simulation. As Abdelmoneim et al. (2020) reported, C-ImmSim plays with the immune response and provided extraordinary analyses, and our results demonstrated the same observations [27]. Despite covering an above-average population coverage, our selected epitopes, VMFMFFYFF, KKMRMAFTFQTMYMG, HQGHRFDQGPSLYLM, QGHRFDQGPSLYLMP, GHRFDQGPSLYLMPK, RFDQGPSLYLMPKYF, QMFNPPFVY, RYCCRRMVL, STIDPAQSY, SIDLNESNKFLAT, SIDLNESNKFLA, QSIDLNESNKFLATADD, NPPFVYSLAISTD, and PFVYSLAISTDGNWI elicited great response owing to their ideal antigenicity with scores averaging between 1.5 and 2.0 and their nonallergenicity. The importance of running protein physicochemical analysis along with verification analyses was understood in the molecular docking process. With the first adjuvant, the docking scores were not impressive because the output protein structure was not stable. However, stabilizing the structure improved the docking analyses 10-fold. These findings on the importance of structure stability were found consistent with Naveed et al. (2021) [25] and Abraham Peele et al. (2020) [28] and improved the quality of the vaccine construct. These results point to a potential agent for the precautionary therapeutics against mucormycosis.

## 4. Materials and Methods

### 4.1. Target Proteins’ Selection

The NCBI protein database was to retrieve 6000 proteins of both *Mucor circinelloides* and *Rhizopus oryzae,* downloaded in FastA format. Gene ontology analysis was run on both the species’ proteins separately using CELLO2GO [29], available at http://cello.life.nctu.edu.tw/cello2go/ (accessed on 23 July 2021). Conservation and homology analyses using Blastp [30] available at https://blast.ncbi.nlm.nih.gov/Blast.cgi?PAGE=Proteins (accessed on 23 July 2021) were performed on both selected proteins individually.

VaxiJen v2.0 [31], available at http://www.ddg-pharmfac.net/vaxijen/VaxiJen/VaxiJen.html (accessed on 23 July 2021), and AllerTOP v.2.0 [32] (https://ddg-pharmfac.net/AllerTOP/ (accessed on 23 July 2021) were utilized for antigenicity and allergenicity analyses of the selected fungal proteins, respectively.

### 4.2. B-Cell Epitope Prediction

B-cell epitopes were predicted using different IEDB tools (http://tools.iedb.org/bcell/ (accessed on 23 July 2021) with plain sequence (no headers) input. BepiPred 2.0 was used to depict linear epitopes based on the random forest algorithm. The surface antigenicity scale put forward by Parker et al. (1986) was dedicated to calculating the surface epitopes of the target proteins [33]. The Kolaskar and Tongaonkar (1990) antigenicity analysis was performed to predict antigenic epitopes in the query proteins [34]. The β-turn prediction allowed the secondary structure analysis based on the target proteins based on their AA sequence. Lastly, hydrophilicity analyses were used to predict the hydrophilic protein regions, which are presumably located on their surface and have a potential antigenic character.

### 4.3. Protein Modeling, Refinement, and Verification

Three-dimensional structures of both selected proteins were modeled using the trRosetta server [35] (https://yanglab.nankai.edu.cn/trRosetta/ (accessed on 23 July 2021). These structures were further utilized to identify the discontinuous B-cell epitopes of the protein discussed in the next section. The models retrieved from trRosetta were refined on GalaxyRefine [36] (http://galaxy.seoklab.org/cgi-bin/submit.cgi?type=REFINE (accessed on 24 July 2021), which were further run on three tools of the UCLA-DOE lab server (https://servicesn.mbi.ucla.edu/PROCHECK/ (accessed on 24 July 2021), PROCHECK [37] ERRAT [38], and verify3D [39] for protein analysis and validation.

### 4.4. Discontinuous B-Cell Epitope Analysis

ElliPro [40] (http://tools.iedb.org/ellipro/ (accessed on 25 July 2021) analyzed the 3D structure of preferable epitopes in both the target proteins. The 3D models of the target proteins (refined PDB formats) were provided to the server, and the parameters were set to default values.

### 4.5. Selection of B-Cell Epitopes

The most probable B-cell epitopes present on the target antigen’s surface were selected for the vaccine construct. Their antigenicity and surface accessibility scores were given prime importance. Furthermore, the final selection of epitopes was carried out by analyzing the allergenicity of all preliminarily selected epitopes through AllerTOP v.2.0 [32] (https://ddg-pharmfac.net/AllerTOP/ (accessed on 26 July 2021) analysis, and the ones with the prediction of being nonallergens were selected for the final vaccine construct.

### 4.6. T-Cell Epitope Analyses

MHC-I [41] and MHC-II [42,43] coverage analyses available on http://tools.iedb.org/main/tcell/ (accessed on 26 July 2021) were performed for potential T-cell epitopes. All the alleles with a length of 9 residues were selected for analysis. Artificial neural network version 4.0 [43] was selected as the prediction method as it is reported to be highly efficient as compared with other predicted models. From the total number of epitopes, only those with IC_50_ values below or equal to 500 were short-listed. Among these, epitopes with acceptable antigenicity and allergenicity were finalized for the construct.

For MHC-II analyses, the prediction method was selected as NN-align-2.2 [42], which analyzes the binding core and affinities, simultaneously providing a robust and adequate prediction. It expels redundant binding cores by rigorous training and provides results for 16 MHC-II classes. Epitopes with the same core peptides and IC_50_ values less than or equal to 100 were selected for further analysis. These were then screened based on their antigenicity and allergenicity predictions.

### 4.7. Population Coverage

All the short-listed epitopes for MHC-I and core epitopes for MHC-II alleles were evaluated for population coverage [44] (http://tools.iedb.org/population/ (accessed on 27 July 2021) before the allergenicity and antigenicity analyses. The ones with higher coverage were further run for these tests.

### 4.8. Selection of T-Cell Epitopes

The most probable T-cell epitopes, present on the target antigen’s surface, were run on AllerTOP v.2.0 [32] (https://ddg-pharmfac.net/AllerTOP/ (accessed on 27 July 2021), and the ones with the prediction of being nonallergens were selected for the final vaccine construct. For MHC-I, the selected peptides were run on VaxiJen v.2.0 (http://www.ddg-pharmfac.net/vaxijen/VaxiJen/VaxiJen.html (accessed on 27 July 2021). For MHC-II, however, this process was further subdivided into two parts. First, the core epitopes were analyzed in a FastA format, and the threshold value was kept at 0.5. All core epitopes having a greater value were subjected to AllerTOP v.2.0 [32], and those predicted as nonallergenic were selected. The epitopes of the selected core peptides were then subjected to VaxiJen v2.0 in the same way as the core peptides. All those with a value greater than 0.5 were further subjected to AllerTOP v.2.0 [32] analysis and finalized for vaccine construct.

### 4.9. In Silico Construction of Vaccine

Potential B-cell and T-cell epitopes for both proteins were further fused into a single chimeric sequence of the vaccine. The final sequence was prepared on a text file. An L7/L12 50S ribosomal protein, following Naveed et al. (2021), was used as an adjuvant at the N-terminal of the vaccine construct as it significantly improves the recognition by Toll-like receptors and can polarize CD4^+^ cells and activate naïve T cells [25]. Three linkers, successfully utilized by Naveed et al. (2021) [25] and Elamin Elhasan et al. (2021) [20], were used to adjoin the epitopes of the two target proteins together. Polyhistidine tags were added at the C-terminal of the vaccine to keep the structure stable.

### 4.10. Antigenicity and Allergenicity Prediction of the Vaccine

The vaccine sequence was subjected to VaxiJen v2.0, Scratch [45], AllerTOP v.2.0 [32], and Allergen FP v.1.0 [32] (http://ddg-pharmfac.net/AllergenFP/data.html (accessed on 28 July 2021) for antigenicity analyses by the first two tools and allergenicity analyses by the last two tools, respectively. Furthermore, since the vaccine construct must not be toxic, the sequence was subjected to ToxinPred analysis [46] available at https://webs.iiitd.edu.in/raghava/toxinpred/algo.php (accessed on 28 July 2021).

### 4.11. Vaccine Secondary Structure Prediction

The primary sequence of the vaccine construct was subjected to secondary structure analysis on PSIPRED [47] (http://bioinf.cs.ucl.ac.uk/psipred/ (accessed on 28 July 2021) and Scratch [45] (http://scratch.proteomics.ics.uci.edu (accessed on 28 July 2021). Sequence data of the vaccine in FastA format were inputted in the required box. The prediction of secondary structure by PSIPRED 4.0, prediction of disordered regions by DISOPRED3 [48], prediction of membrane helices by MEMSTAT-SVM [49], prediction of domains by DomPred [50], 3D structure prediction based on domain folding by DMPfold 1.0 [51], and contact prediction by DeepMetalPSICOV 1.0 [52] was analyzed using the server. For scratch-assisted prediction, a plain sequence of the vaccine with no headers was provided to the server, and its secondary structure, disordered regions, solvent accessibility, disulfide bonds, and antigenicity were predicted.

### 4.12. Vaccine Tertiary Structure Prediction, Refinement, and Verification

The 3D structure of the vaccine was modeled using the trRosetta server [35] (https://yanglab.nankai.edu.cn/trRosetta/ (accessed on 28 July 2021). The model retrieved from trRosetta was refined on GalaxyRefine [36] (http://galaxy.seoklab.org/cgi-bin/submit.cgi?type=REFINE (accessed on 29 July 2021) and was subjected to three verification tools of the UCLA-DOE lab server (https://servicesn.mbi.ucla.edu/PROCHECK/ (accessed on 29 July 2021), PROCHECK [37], ERRAT [38], and verify3D [39]. The Z-Lab RC plot server [53] (https://zlab.umassmed.edu/bu/rama/ (accessed on 29 July 2021) was also utilized for an additional confirmatory RC analysis.

### 4.13. Physicochemical Properties

The ProtParam tool (https://web.expasy.org/protparam/ (accessed on 29 July 2021) of the Expasy server [54] was utilized to examine the physicochemical characteristics of the vaccine construct. The parameters computed for the vaccine include molecular weight, stability index, estimated half-life, and GRAVY (the more positive the value, the more hydrophilic the structure is). Further, Protein-Sol [55] (https://protein-sol.manchester.ac.uk (accessed on 29 July 2021) was used to analyze the solubility index of the protein compared with a standard *E. coli* protein.

### 4.14. ElliPro and Cleavage Analysis of the Vaccine

The vaccine construct was subjected to ElliPro [40] and MHC-NP cleavage analyses [43] (http://tools.iedb.org/mhcnp/ (accessed on 30 July 2021) to check for the conservation of discontinuous B-cell epitopes and T-cell epitopes. The MHC-NP tool analyzed the epitopes processed by the MHC molecules upon interaction. All alleles of the human species were selected to run the analysis. The protein sequence was provided in FastA format, and the default values were not changed.

### 4.15. Molecular Docking of Vaccine with Immune and General Receptors

To analyze whether the designed construct binds with the general mucormycosis and immune receptors, crystal structures of three receptors, MHC-I (PDB ID: 2XPG), TLR2 (PDB ID: 6NIG), and GRP78 (PDB ID: 5F1X), were retrieved from the Protein Data Bank [56] (https://www.rcsb.org (accessed on 30 July 2021). MHC-1 interaction with the vaccine is crucial as it represents the antigen to the cytotoxic T cells. TLRs further play a role in recognizing specific PRRs that initiate an innate immune response. GRP78 plays a role as a general attachment point for mucormycosis agents, and interaction with GRP78 confirms the conservation of these epitopes and their recognition as fungal entrants.

The docking analyses of the vaccine along with three individual receptors were run on ClusPro supercomputers [57] (https://cluspro.bu.edu/home.php (accessed on 30 July 2021). The best model clusters were analyzed based on model scores and binding affinities (negative energies were the standard) and were further processed for binding affinity analyses. PyMOL2 [58] downloaded from https://pymol.org/2/ (accessed on 30 July 2021) and Discovery Studio Visualizer downloaded from https://discover.3ds.com/discovery-studio-visualizer-download (accessed on 30 July 2021) were utilized to visualize the binding residues between the vaccine and the receptors. The MDWeb tool [59] (https://mmb.irbbarcelona.org/MDWeb (accessed on 30 July 2021) was utilized for the simulations of the docked complexes.

### 4.16. Codon Optimization

Reverse translation of our vaccine construct was retrieved from EMBOSS Backtranseq (https://www.ebi.ac.uk/Tools/st/emboss_backtranseq/ (accessed on 31 July 2021) using an *E. coli* reverse translation codon table as the vaccine was expressed in its vector. The input sequence was provided in plain one-letter-residue format. After reverse translation, the nucleotide sequence was pasted on the JAVA codon adaptation tool [60] (http://www.jcat.de/ (accessed on 31 July 2021), and codon optimization was performed to adapt to most sequenced eukaryotic and prokaryotic organisms. The *E. coli* strain K-12 was utilized for the analysis.

### 4.17. Cloning and Expression Analysis

An *E. coli* pBR322 expression vector was utilized from https://www.snapgene.com/resources/plasmidfiles/?set=basic_cloning_vectorsandplasmid=pBR322 (accessed on 31 July 2021), as it is a standard in cloning procedures and is optimum for quality results. SnapGene software downloaded from https://www/snapgene.com/ (accessed on 31 July 2021) was used to perform the cloning and expression. The insert method was used for this purpose.

### 4.18. Immune Simulations

To verify the immune response generated by the vaccine construct, the sequence was run on the C-ImmSim server [61] (http://kraken.iac.rm.cnr.it/C-IMMSIM/ (accessed on 1 August 2021). The server provides immune activity against vaccines or drugs based on particulars, such as injection time. It shows B and T lymphocytes’ response to the vaccine along with the prediction of immunoglobulins’ and immunocomplexes’ response.

## 5. Conclusions

This study proposed a potential vaccine against mucormycosis, one of the daunting challenges faced by medical professionals due to immunosuppression of COVID-19-affected patients. Using the immunoinformatics approach and immune simulations, epitopes from two proteins of different species were selected. It was found that several epitopes of both proteins fit well in the preferred regions, and after analyzing the population coverage and immune response simulations, it was concluded that the designed construct, if manufactured industrially, holds great promise. The vaccine structure was found to be stable with only 7% disordered regions. Docking analyses with two immune receptors and one natural fungal receptor demonstrated great potential with a variety of interactions. Lastly, this study eliminates the economic and emotional burden of physical screening pre- and, to some extent, postmanufacture, easing and enhancing the whole vaccine design process.

## Figures and Tables

**Figure 1 vaccines-10-00664-f001:**
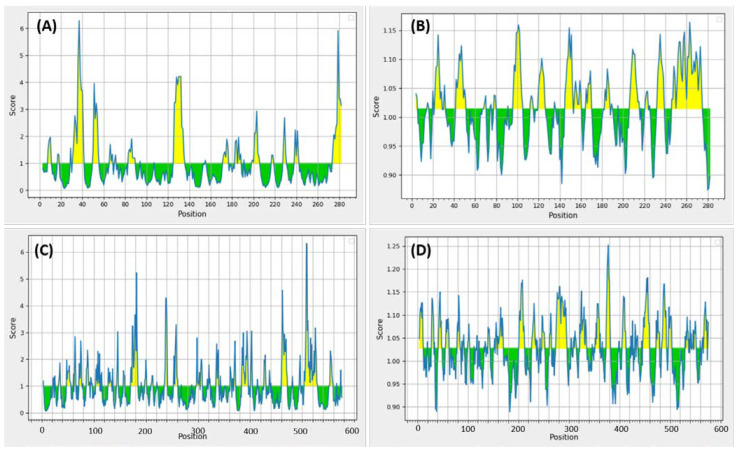
(**A**) Antigenicity scores of phytoene dehydrogenase residues. (**B**) Antigenicity scores of the hypothetical protein residues. (**C**) Surface accessibility scores for the residues of phytoene dehydrogenase. (**D**) Surface accessibility scores for the residues of the hypothetical protein.

**Figure 2 vaccines-10-00664-f002:**
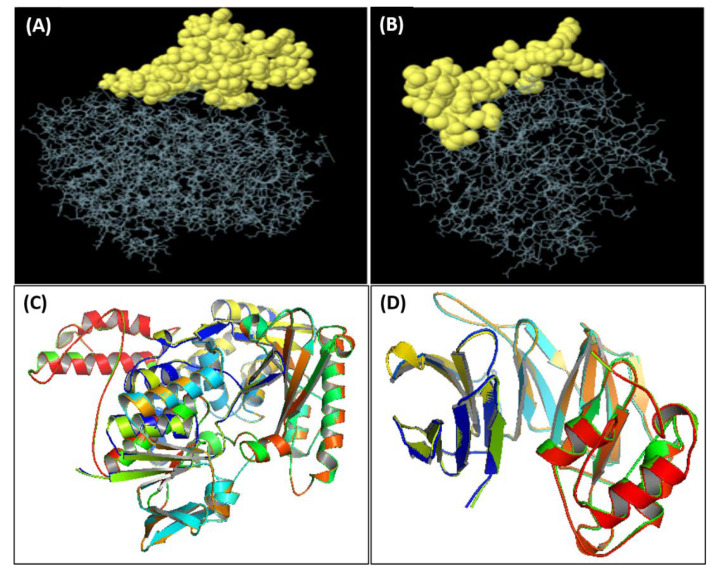
(**A**) Discontinuous epitopes of phytoene dehydrogenase, (**B**) discontinuous epitopes of the hypothetical protein, (**C**) superimposed raw and refined models of phytoene dehydrogenase, (**D**) superimposed raw and refined models of the hypothetical protein.

**Figure 3 vaccines-10-00664-f003:**
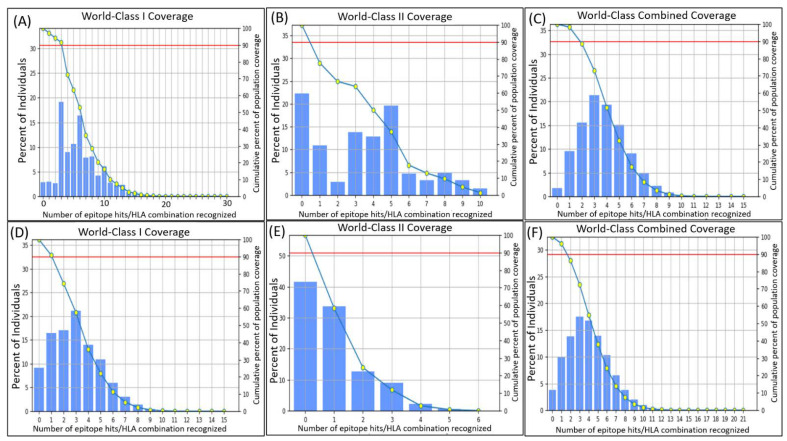
(**A**) World coverage of phytoene dehydrogenase’s selected epitopes for MHC-I class. (**B**) World coverage of phytoene dehydrogenase’s selected epitopes for MHC-II class. (**C**) World coverage of selected epitopes of phytoene dehydrogenase for both MHC classes. (**D**) World coverage of hypothetical protein’s selected epitopes for MHC-I class. (**E**) World coverage of hypothetical protein’s selected epitopes for MHC-II class. (**F**) World coverage of hypothetical protein’s selected epitopes for both MHC classes.

**Figure 4 vaccines-10-00664-f004:**
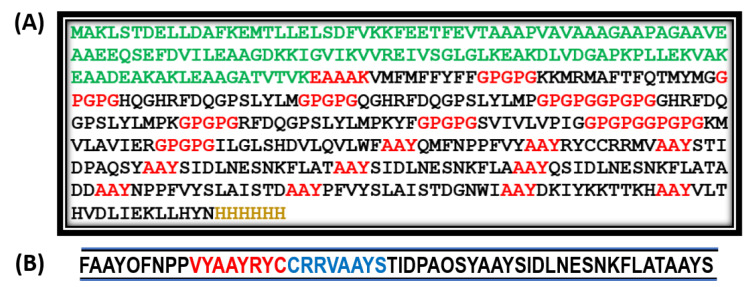
(**A**) Final vaccine construct of all antigenic but nonallergenic epitopes selected from the target proteins; (**B**) ToxinPred predicting the toxic residues in red and potential toxic residues in blue.

**Figure 5 vaccines-10-00664-f005:**
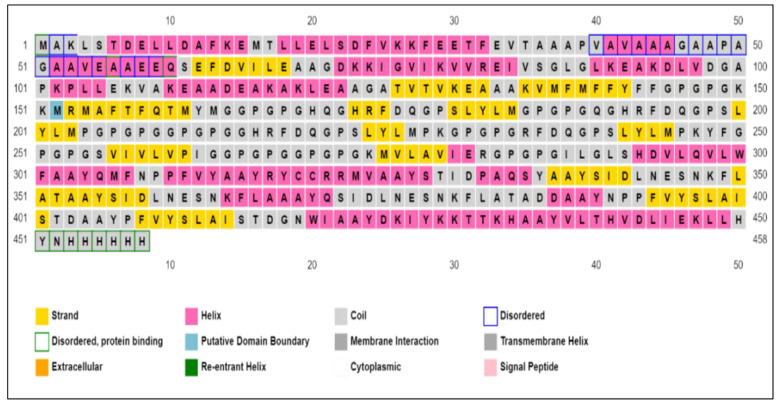
Secondary structure prediction of the constructed vaccine.

**Figure 6 vaccines-10-00664-f006:**
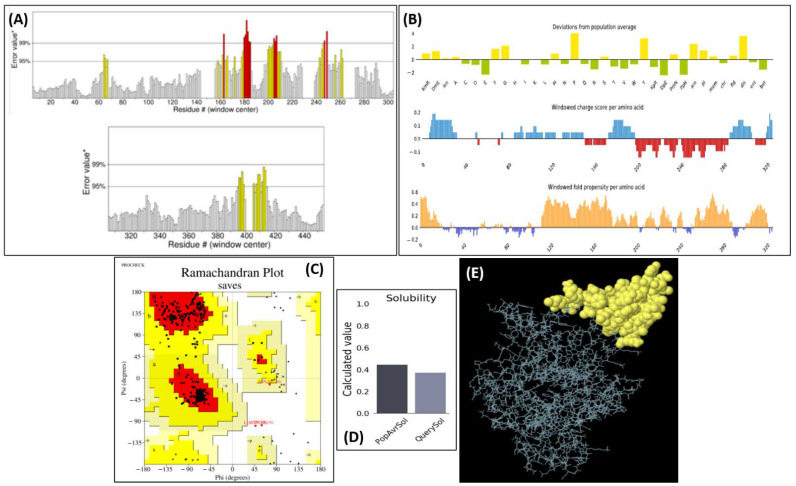
Physicochemical properties of the vaccine; (**A**) ERRAT score of the vaccine construct, (**B**) RC plot of the vaccine construct predicting the secondary structure, (**C**) vaccine solubility as compared with the reference *E. coli* solubility index, (**D**) tertiary structure of the vaccine’s discontinuous epitopes, and (**E**) solubility index along with fold propensity for the vaccine construct.

**Figure 7 vaccines-10-00664-f007:**
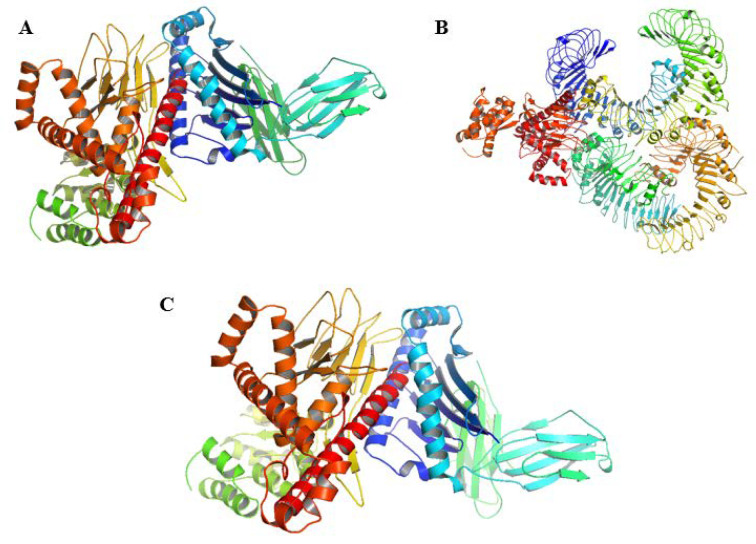
(**A**) Docking complex of MHC-1 with the vaccine, (**B**) docking complex of TLR2 with the vaccine, (**C**) docking complex of GRP78 with the vaccine.

**Figure 8 vaccines-10-00664-f008:**
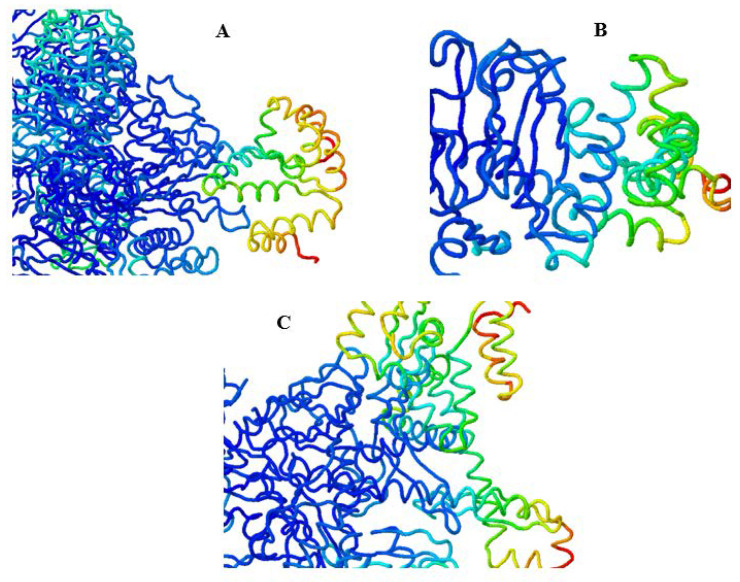
(**A**) MD simulations of vaccine and GRP78, (**B**) MD simulations of vaccine and MHC-I, (**C**) MD simulations of vaccine and TLR2.

**Figure 9 vaccines-10-00664-f009:**
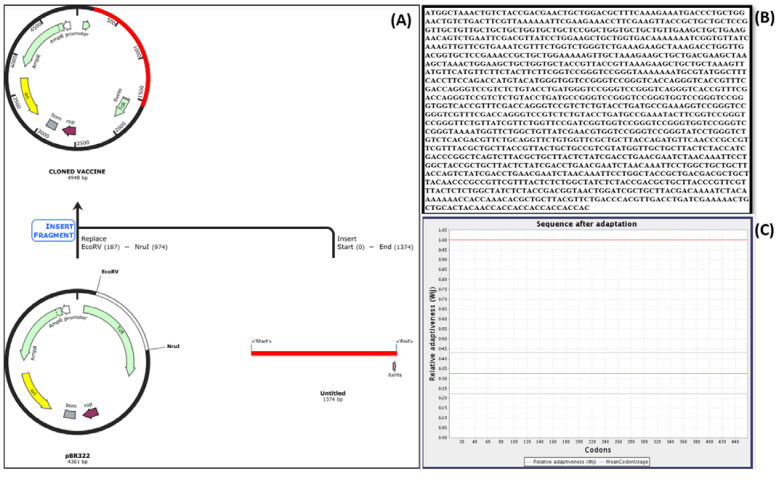
(**A**) Cloning process of vaccine into pBR322, (**B**) refined sequence according to codon optimization, (**C**) codon adaptation postoptimization.

**Figure 10 vaccines-10-00664-f010:**
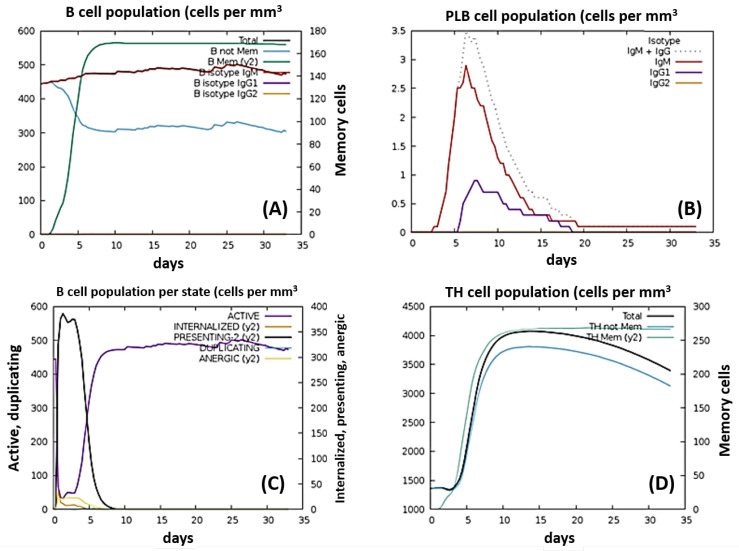
B- and T-cell populations upon vaccine injection: (**A**) antibody isotopes produced and activated in response to the vaccine, (**B**) PLB cells’ response against vaccine, (**C**) B-cell response per state, (**D**) helper T-cell response against the vaccine.

**Table 1 vaccines-10-00664-t001:** Antigenicity score of the selected proteins using VaxiJen v2.0.

Protein	Antigenicity Score
Phytoene dehydrogenase	0.5642
Hypothetical protein	0.9140

**Table 2 vaccines-10-00664-t002:** Finalized B-cell epitopes for the vaccine construct.

No.	Start	End	Peptide	Length	Antigenicity	Surface Accessibility	Allergenicity
**Phytoene dehydrogenase**							
1	374	382	SVIVLVPIG	9	0.9144	1	Nonallergen
2	405	413	KMVLAVIER	9	1.1837	1	Nonallergen
3	446	459	ILGLSHDVLQVLWF	14	0.9727	1	Nonallergen
**Hypothetical protein**							
1	31	41	DKIYKKTTKH	10	1.055	6.293	Nonallergen
2	260	274	VLTHVDLIEKLLHYN	15	1.164	2.059	Nonallergen
3	120	135	IQLISPPSKKSKTT	14	1.2688	4.217	Nonallergen

**Table 3 vaccines-10-00664-t003:** World population coverage of MHC-I and MHC-II epitopes of the selected proteins.

Phytoene Dehydrogenase	
Epitope	Coverage
**MHC-I coverage**	
AAFWVMFMF	27.24%
KIYDRASKY	41.99%
SSISFYWSM	27.64%
MAFTFQTMY	62.52%
STFPVWFWL	53.56%
VMFMFFYFF	68.08%
LTSSSISFY	64.02%
LVYAYHNILL	59.37%
RMAFTFQTM	37.16%
WVMFMFFYF	50.11%
**MHC-II coverage**	
FDQGPSLYL	44.03%
FIYNAPVAK	16.02%
FKTKKMRMA	22.39%
YMGMSPYDA	18.55%
YFKTKKMRM	36.85%
MRMAFTFQT	60.71%
LRCDNNYKV	6.69%
LAVIERRLG	31.26%
FYWSMSTKV	33.61%
FYVNVPSRI	43.45%
**Hypothetical Protein**	
**MHC-I coverage**	
QMFNPPFVY	66.02%
KVYEWDFSR	45.05%
RYCCRRMVL	49.95%
STIDPAQSY	45.26%
YLSLLQAEY	35.72%
**MHC-II coverage**	
AQMFNPPFV	8.91%
FVYSLAIST	26.79%
IDLNESNKF	33.99%
IDPAQSYQL	17.55%
IEKLLHYNP	7.01%

**Table 4 vaccines-10-00664-t004:** Selected T-cell epitopes for vaccine construct and their antigenicity scores.

Epitope	Antigenicity Score
**Phytoene dehydrogenase**	
VMFMFFYFF	0.6814
KKMRMAFTFQTMYMG	0.5753
HQGHRFDQGPSLYLM	0.9144
QGHRFDQGPSLYLMP	1.3727
GHRFDQGPSLYLMPK	1.2474
RFDQGPSLYLMPKYF	1.8275
**Hypothetical protein**	
QMFNPPFVY	0.582
RYCCRRMVL	2.4632
STIDPAQSY	0.8302
SIDLNESNKFLAT	1.6163
SIDLNESNKFLA	1.5324
QSIDLNESNKFLATADD	1.3854
NPPFVYSLAISTD	2.0289
PFVYSLAISTDGNWI	1.8247

**Table 5 vaccines-10-00664-t005:** Antigenicity and allergenicity analyses of the vaccine construct.

Tool Used	VaxiJen v. 2.0	Scratch	AllerTOP v.2.0	AllergenFP v. 1.0
Score	0.7572 (antigenic)	0.5 (antigenic)	Non-allergen	Nonallergen

## Data Availability

The data set used in the current study will be made available at reasonable request.

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
