# Peer review of "A Vaccine Construction against COVID-19-Associated Mucormycosis Contrived with Immunoinformatics-Based Scavenging of Potential Mucoralean Epitopes"

_vaccines, 2022, doi:10.3390/vaccines10050664_

Round 1

Reviewer 1 Report

Line 26 Fix the following sentence fragment: „Abstract: A higher taper of mechanical preparation offers sufficient enlargement of the r Abstract…“

Line 54 add a citation for the sentence: Mucormycosis has plagued the world for five decades, but little is known about its causative agents, pathogenesis, and epidemiology.

Line 65 DM = ?

Line 70 ->first the description and then the abbreviation

Figure 1 improve X axes of C and D

Figure 3 enhance the quality of image

Figure 4 B enhance the quality of image

Figure 5 enhance the quality of image

Figure 10 enhance the quality of image, position the label under the curves

Author Response

We would like to thank the academic editor and the reviewers for taking out their precious time to review this manuscript and give us their comments. We would like to explicitly state that we agree with all the comments as these helped us improve the quality of our paper. We have made a conscious effort to answer all the remarks in the paper as advised by the reviewers and highlighted changes made in yellow for their convenience. Kindly consider these and excuse us for any lapse on our part.

Reviewer 2 Report

In this manuscript, Naveed et al. describe the use of bioinformatic tools, B- and T-cell epitope prediction, in silico vaccinology, and immunoinformatics to derive a proposed vaccine for the prevention of COVID-19-associated mucormycosis, a potentially fatal fungal infection particularly in immunocompromised hosts. The authors selected phytoene dehydrogenase from a Mucor species and a hypothetical protein from a Rhizopus species as the vaccine targets based on their transmembrane location (surface accessibility), high conservation among Mucor and Rhizopus members, predicted antigenicity, and non-allergenicity. Using an immunoinformatic approach, they found that their proposed vaccine construct would elicit a strong predicted humoral and cell-mediated immune response, showing promise as a potential anti-mucormycosis vaccine. While none of the results reported here are supported by in vitro and/or in vivo evidence, this study does illustrate the utility of computational and immunoinformatic approaches in the early developmental phases of vaccine construction and efficacy assessment. Overall, the writing in this manuscript is clear and understandable, and the experimental approach is sound. I do recommend that the authors use an experienced proofreader in English to resolve some of the mistakes in expression and grammar. My specific comments/suggestions are provided below.

  • Abstract (line 26): Please remove the line starting with “Abstract: A higher taper of mechanical preparation….” This does not seem to belong to this manuscript.
  • Abstract (line 40): For accuracy, please revise “It elicited a tremendous innate immune response…” to “It elicited a strong predicted innate and adaptive immune response….”
  • Results (line 114): Please change “was” to “were” to correct the grammatical mistake.
  • Results (line 116): Is there a computational score associated with determining whether a protein is non-allergenic?  If so, why weren’t these scores included along with the antigenicity scores for the phytoene dehydrogenase and hypothetical protein?
  • Results (line 131): Please discuss Figure 1(A) and (B) in the Results section and specify the difference between these two graphs.  Also, specify the differences between Figure 1(A) and (B) in the figure legend.
  • Results (lines 173-182): I suggest listing the linear and discontinuous epitopes for phytoene dehydrogenase and the hypothetical protein in a supplementary table.
  • Results (line 188): Please change “base” to “basis”.
  • Results: Please clearly distinguish between the graphical representation of coverage (Figure 3) and Table 3 in which epitope 3 (MAFTFQTMY) has a coverage of 62.52%. It is confusing as written.
  • Results (lines 249 and 250): Why were epitopes KVYEWDFSR and YLSLLQAEY excluded from the list showing the best individual coverages?  MHC-I coverage for these epitopes was higher than the MHC-II coverage for IDLNESNKF.
  • Results (line 246): Please replace the word “brilliant” with a more precise term.
  • Results: Resolution of Figure 4(B) is poor. Please improve the quality of this figure.
  • Results: For Figure 10, panels A and C, the legend interferes with the curves in the graph. Please fix this, so that the legend and curve do not obstruct one another.         

Author Response

(The authors gave the same response as above.)
